# A Portable Device for the Measurement of Venous Pulse Wave Velocity

Agata Barbagini [1] , Leonardo Ermini [1,*] , Raffaele Pertusio [1] , Carlo Ferraresi [2] and Silvestro Roatta [1]

1 Laboratory of Integrative Physiology, Department of Neuroscience, University of Torino, C.so Raffaello 30, 10125 Torino, Italy; agata.barbagini@unito.it (A.B.); raffaele.pertusio@unito.it (R.P.); silvestro.roatta@unito.it (S.R.)

2 Department of Mechanical and Aerospace Engineering, Politecnico of Torino, C.so Duca degli Abruzzi 24, 10129 Torino, Italy; carlo.ferraresi@polito.it

* Correspondence: leonardo.ermini@unito.it

**Abstract:** Pulse wave velocity in veins (vPWV) has recently been reconsidered as a potential index of vascular filling, which may be valuable in the clinic for fluid therapy. The measurement requires that an exogenous pressure pulse is generated in the venous blood stream by external pneumatic compression. To obtain optimal measure repeatability, the compression is delivered synchronously with the heart and respiratory activity. We present a portable prototype for the assessment of vPWV based on the PC board Raspberry Pi and equipped with an A/D board. It acquires respiratory and ECG signals, and the Doppler shift from the ultrasound monitoring of blood velocity from the relevant vein, drives the pneumatic cuff inflation, and returns multiple measurements of vPWV. The device was tested on four healthy volunteers (2 males, 2 females, age $33 \pm 13$ years), subjected to the passive leg raising (PLR) manoeuvre simulating a transient increase in blood volume. Measurement of vPWV in the basilic vein exhibited a low coefficient of variation ($3.6 \pm 1.1\%$), a significant increase during PLR in all subjects, which is consistent with previous findings. This device allows for carrying out investigations in hospital wards on different patient populations as necessary to assess the actual clinical potential of vPWV.

**Keywords:** Raspberry Pi; pulse wave velocity; volume status; vascular stiffness; venous compliance; bioengineering

## 1. Introduction

Nowadays, the estimation of volaemic status (or volaemia) of a patient through a noninvasive method is not trivial but its development could be of great utility, given that interest in the patient's volaemic spans across several different hospital departments [1]. An innovative method for assessing the volaemic status of patients has recently been proposed: pulse wave velocity in veins, termed venous pulse wave velocity (vPWV) [2,3]. The relation among vPWV, venous pressure, and changes in blood volume was first observed in the 1970s [4,5], but the methodology was subsequently abandoned, possibly due to the difficulty of achieving reliable measurements. In fact, pulse wave velocity is commonly measured in the arteries and widely used as a marker of cardiovascular risk [6]. Both for arteries and veins, PWV is related with the elasticity of the vessel. This dependence is formalized by the Moens–Korteweg equation, according to which the value of PWV is proportional to the square root of the incremental elastic modulus of the vessel wall:

$$\text{PWV} = \sqrt{\frac{E_{inc} \times h}{2r\rho}} \qquad (1)$$

where $E_{inc}$ represents the incremental elastic modulus of the vessel wall, $h$ is the wall thickness, $r$ is the vessel radius, and $\rho$ is the blood density. The incremental elastic modulus

is proportional to the filling of the vessel. In case of veins, their filling should be a good proxy of the volaemic status of the subject, with veins being the capacitance vessels that are used as a blood volume reservoir [2].

PWV measurement is considerably easier in the arteries than in the veins due to higher blood pressure in the arteries and the presence of a natural cardiac pulsatility that is absent in veins. However, in recent studies, an improved methodology was devised in which a pneumatic compression applied to a limb extremity (e.g., the hand), generates the pressure pulse (i.e., the pulse wave) in the venous compartment, which centrally propagates and is proximally detected (e.g., at the basilic vein) by Doppler ultrasound. Knowing the distance between cuff and ultrasound probe, it is possible to calculate the vPWV. In addition, to limit the confounding effect of respiratory and cardiac modulation of venous blood pressure, compression is delivered synchronously with cardiac and respiratory activity. Indeed, venous flow has great intersubject variability. In some subjects, backward pulsatility related to right atrium contractions can be observed; in others flow, is almost constant. Sending an exogenous compression synchronised with cardiac and respiratory activity allows for the measurement to be applicable and repeatable over a large pool of subjects, regardless of whether they have inherent pulsatility [2,3]. It was thus possible to measure vPWV with reasonable reliability and to confirm linear correlation between vPWV and venous pressure [2]. vPWV also exhibited good sensitivity to simulated changes in blood volume, associated with minor changes in venous pressure as observed in healthy subjects in response to passive leg raising (PLR) [3]. These results were also achieved thanks to the development of an automatic method for calculating vPWV by analysing the Doppler shift signal, as acquired by the echograph machine [2]. Thanks to this feature, the measurement can be carried out continuously and is thus adequate for long-term monitoring.

Following these promising results in healthy subjects, clinical trials need to be carried out in order to assess the usefulness of vPWV as a diagnostic haemodynamic parameter. However, experimental series in hospital wards are not yet feasible due to cumbersome laboratory instrumentation and no dedicated device being available on the market. The aim of this study is to present a proof-of-concept prototype of a user-friendly, portable, and electrically isolated device for vPWV measurement. The entire hardware of the device is embedded in a small, handy box thanks to a single board computer, the Raspberry Pi, wirelessly connected to a PC, where a graphic interface allows for interacting with the device. The performance of the RPi-based device was tested on 4 healthy subjects subjected to PLR, and it is compared with measurements performed by the original laboratory equipment (PC-based system).

## 2. Materials and Methods

The devised methodology for vPWV measurement requires the acquisition of the following signals:

- respiratory signal;
- ECG; and
- Doppler shift.

The flowchart of measurement process is reported in Figure 1, while a schematic representation of the device is shown in Figure 2. Examples of acquired signals are reported in Figure 3, showing typical timing of acquisition. Signals were acquired one at a time to ensure the highest and most stable sampling rate. Raspberry Pi controls the timing and switches the signal acquisition, following the order shown in the list above. The delay introduced by switching the acquisition from one channel to the next was measured and found to be negligible (<2 ms). In order to synchronize the delivery of the compressive stimulus both with respiratory and cardiac cycles, the system first identifies the expiratory phase in the respiratory signal and then the rising edge of the R-wave in the ECG signal. This latter event triggers the delivery of the compressive stimulus, after an optional delay defined by the user, and the acquisition of the haemodynamic response (i.e., the Doppler shift). This signal is then processed to identify the footprint that corresponds to the passage

of the pulse wave under the probe. In this way. the transit time of the pulse wave (i.e., the pulse transit time) can be estimated and its velocity calculated on the basis of cuff-probe distance.

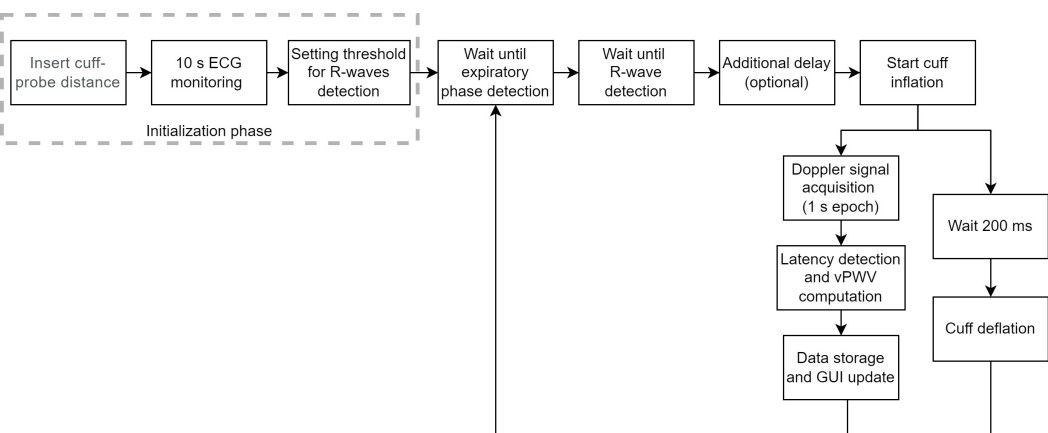

**Figure 1.** Flowchart of measurement. Compressive stimulus synchronized with respiratory and cardiac cycle. PW transition under Doppler probe allows for calculating vPWV.

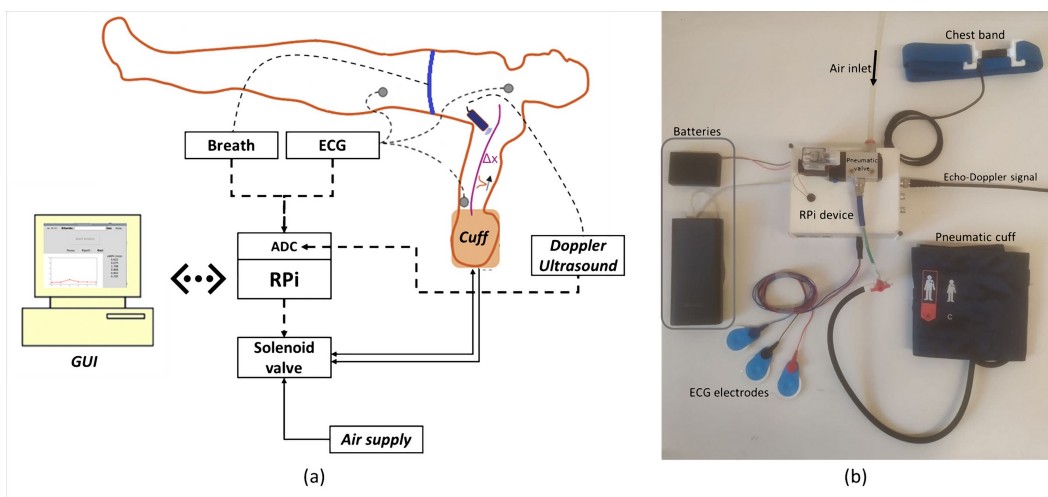

**Figure 2.** Device schematic and photographic representation. (**a**) Functional scheme. Electrical and pneumatic connections indicated by dashed and solid lines, respectively. Raspberry Pi (RPi) is the control unit wirelessly connected to a graphical computer interface on a PC. (**b**) Picture of the prototype enclosed in a 3D-printed box and connected to the external batteries, ECG electrodes, chest band, and Doppler signal (ultrasound machine not shown). Pneumatic valve was mounted on top of the device, and connected to compressed air supply (not shown) and cuff.

### 2.1. Hardware

PC board Raspberry Pi 4 B+ (processor: Broadcom BCM2711 quadcore 64 bit ARM Cortex-A72, 1.5 GHz; RAM: 4 GB) was chosen because of its small size, low cost, and ability to be programmed using Python, which allows for efficient signal processing. In addition, this solution offers easy design of graphical interface, the possibility of saving data, access to a large range of open-source libraries, and an easy connection to a remote PC. Alternative solutions based on microcontrollers do not provide all these possibilities. An analogue-to-digital (A/D) converter was used in combination with Raspberry Pi to acquire biological signals, exploiting the serial peripheral interface (SPI) communication protocol: the Waveshare Raspberry Pi High-Precision AD/DA Expansion Board (Waveshare Electronics, Shenzhen, China) can be easily assembled on top of Raspberry Pi. It embeds Texas Instruments' ADS1256 chip and provides eight channels for the 24-bit single-ended acqui-

sition of analogue signals, while keeping free the RPi's 40 general-purpose input/output (GPIO) pins. The hardware also included printed circuits boards for conditioning respiratory signal, a relay, and a pneumatic valve. The electrical power source is external, consists of two batteries (5 V, 3A ), and through the DC/DC converter, it supplies:

- Raspberry Pi (5 V, 2 A);
- pneumatic valve (12 V, 200 mA);
- breathing signal conditioning circuit (12 V, 200 mA).

Other components requiring power supply, such as the relay and the ECG board, were connected to the Raspberry Pi's output pins, each of which could supply either 5 or 3.3 V (max 16 mA). As such, the device has a floating ground and is electrically safe.

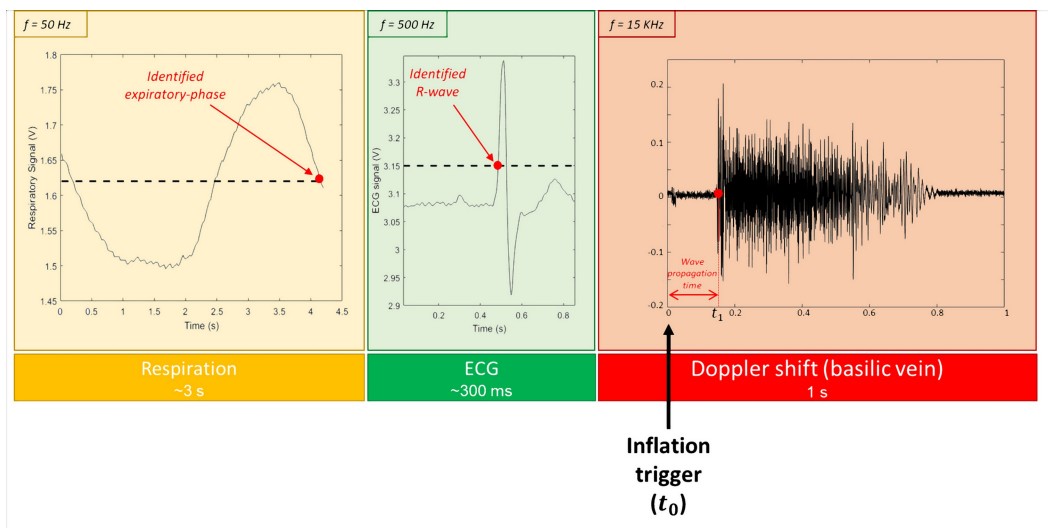

**Figure 3.** Signal acquisition timeline, typical timing, and signal waveforms. (right) Gaemodynamic response to compressive stimulus reported in time domain. Compressive stimulus delivered at time 0 of Doppler acquisition, and pulse transit time $t_1$ could be easily estimated.

### 2.2. Respiratory Signal

A custom chest band (3 cm wide) produced of elastic material, and integrating a resistive sensor (strain gauge), is used to monitor respiratory movement. Before being digitally sampled (50 Hz), the respiratory signal is high-pass filtered (cut off frequency: 0.1 Hz) to remove slow trends in signal. It is offset to about 1.6 V by a summing circuit, to fit the input range of the acquisition board (0–5 V). RPi identifies the expiratory phase when the signal drops below the average value (about 1.6 V), as shown in Figure 3.

### 2.3. ECG

The ECG is provided by an integrated signal conditioning block: AD8232 SparkFun Single Lead Heart Rate Monitor (SparkFun, Niwot, Colorado). It presents a jack plug connection for three ECG electrodes (two measuring electrodes and one reference), and yields a fully positive, amplified, and low-pass filtered analogue output that can be directly fed to the acquisition board and sampled at 500 Hz. ECG was monitored for 10 seconds at the beginning of the experimental session (initialization phase) in order to estimate an appropriate threshold for R-wave detection: the threshold was initially set at 15% of the baseline-to-peak amplitude. It was subsequently raised if an excessive number of peaks were crossed within the recorded 10 seconds. R-waves waves were then automatically detected by the upward crossing of the threshold, as shown in Figure 3. AD8232 needs to be powered at 3.3 V, which was provided by one of RPi's output pins.

### 2.4. Compressive Stimulus

Immediately after the detection of the R-wave, Raspberry Pi triggers the compressive stimulus. The command is imparted through a digital output (3.3 V) which activates a relay (KY-019 5V Relais module) which, in turn, drives a solenoid valve (3V210-08 12VDC 3-Way, Heschen, Zhejiang, China), connecting a compressed air supply (2 bar), to the cuff (49 × 15 cm, GIMA, Gessate, Italy) wrapped around the wrist. Inflation lasts for 200 ms, so that the cuff, which initially is deflated, reaches a peak pressure in the range of 150–250 mmHg. After that, RPi lowers the digital output, which releases the valve so the cuff passively deflates.

### 2.5. Ultrasound Doppler Shift Signal

The Doppler shift signal is generally provided by ultrasound machines as an audio signal output (e.g., from the headphone output). In the present study, a MyLab25 Gold (ESAOTE, Genova, Italy) equipped with a linear probe (LA523, ESAOTE, Genova, Italy) was employed. The probe was located transversally over the medial side of the arm, about 15 cm proximal to the elbow, with an angle of about 60 degrees to the basilic vein (BV). Using a pulsed-wave Doppler, it is possible to locate the sample volume precisely over the basilic vein. The Doppler shift is fed to the acquisition board (sampling frequency: 7.5 kHz). The acquisition of a 1 s epoch starts immediately after the trigger of the compressive stimulus. After completing the acquisition, the Doppler shift was processed in the time domain as follows. The root mean square envelope of the signal is extracted. The latter is smoothed through a moving average filter (sliding square window of 100 ms). The footprint was then identified as instant $t_1$ at which the envelope reaches 5% of its baseline-to-peak amplitude. The latency (i.e., the pulse transit time) is then estimated as $\Delta t = t_1 - t_0$ (where $t_0$ corresponds to the instant when the trigger is delivered, equal to zero in Figure 3), and the vPWV can lastly be calculated as $\frac{\Delta x}{\Delta t}$.

### 2.6. Graphical Interface

A graphical interface was developed to interact with the device. Raspberry Pi was configured as an access point, so it offers a Wi-Fi network that allows for a remote PC to connect and interact with the graphical interface while avoiding any electrical connection to the PC. The GUI shows the vPWV values in graphical and numerical form, periodically updating the display at each new measurement. Moreover, it allows for the measurement process to be paused and resumed at any time after the initialisation phase. The user is asked to insert the $\Delta x$ cuff-probe distance value at beginning of the process, necessary for vPWV computation. The user can also adjust the delay of the compressive stimulus with respect to R-wave detection. This feature can be exploited to optimize the timing of the pressure pulse generation with respect to the spontaneous fluctuations of venous blood flow (of cardiac origin) which may disturb the detection of the pulse wave footprint in the Doppler shift. All acquired signals and vPWV values are locally stored and can be transferred to the PC at the end of the session, for further processing. An example of the GUI layout is shown in Figure 4.

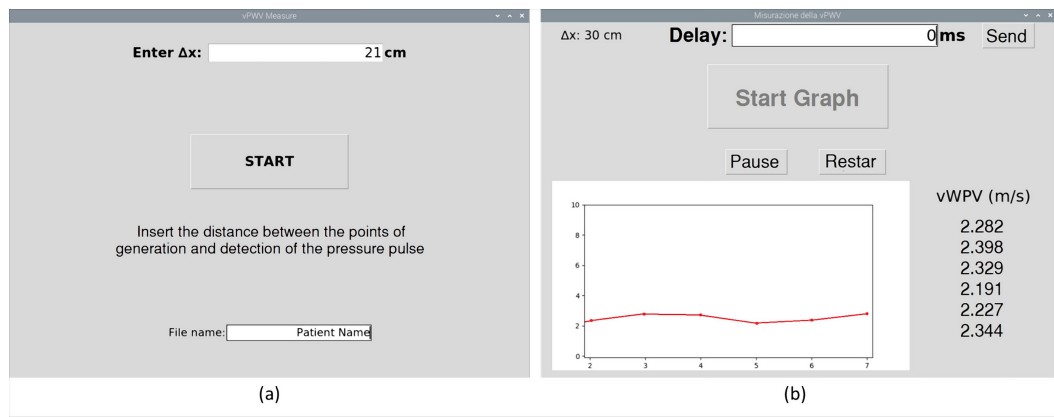

**Figure 4.** Graphical user interface. (**a**) GUI's first page allows to enter the preliminary information, such as the distance between the cuff and the ultrasound probe ($\Delta x$), and the name for saving the data. (**b**) In working mode, the GUI displays numerical and graphical representation of measured vPWV values, and allows to adjust the additional delay for the delivery of the the compressive stimulus.

## 3. Device Validation

### 3.1. Procedure

In order to verify the performance of the device, vPWV measurements were carried out on four healthy subjects (two males and two females, age $33 \pm 13$ years). Each subject provided informed consent, and the experiment was approved by the Ethics Committee of the University of Turin (23 March 2015) in accordance with the guidelines of the Declaration of Helsinki. Tests were performed during resting condition, in supine position, and during a simulated increase in blood volume in the trunk, as obtained by PLR. Each subject lay in supine position for at least 30 minutes before starting the measurement in order to achieve a stable haemodynamic condition [7]. The cuff for the delivery of the pneumatic compressive stimuli was placed around the wrist, and the Doppler ultrasound probe was placed over the BV about 15 cm proximally to the elbow. An electric compressor supplied the high-pressure air. Respiratory movements and ECG were also collected and all transducers and devices were connected to the prototype as described above. In addition, all signals were also continuously sampled by an external acquisition board (Micro 1401 IImk, CED, Cambridge, UK, with Spike2 software) and off-line processed in Matlab®, according to the original methodology [3]. After the initialisation phase, 10 vPWV values were collected, while the subject remained at rest in supine position (baseline); then, the PLR was performed by an operator manually raising the subject's leg at about 45 degrees. Then, 5 to 7 vPWV readings were taken in this position, and another 5 to 10 readings after returning to supine position (post-PLR). A time interval of 15–30 s elapsed between subsequent measurements. The whole session lasted about 45 min.

### 3.2. Data Analysis

After the acquisition with the RPi-based device, the data were cleaned by removing values acquired during the movement of the ultrasound probe or a lack of contact with the arm. The reliability of the measurement was assessed by the coefficient of variation (CoV = STD/MEAN $\times$ 100), calculated over the 10 vPWV values collected during rest in the supine position. Afterwards, the impact of PLR on vPWV was investigated. The basal value was set as the average of the values relative to the supine position. vPWV collected during PLR were expressed as a percent changes with respect to the baseline. Moreover, the sensitivity of the measurement to simulated blood volume changes was assessed by individually comparing data collected during baseline, PLR and post-PLR, by Student's t-test with Bonferroni correction for multiple comparisons. Lastly, measurement accuracy was assessed by comparing each vPWV value yielded by the prototype with the

corresponding value measured by the original PC-based system. The difference among them was expressed in percentage terms with respect to the mean of the two values.

## 4. Results

In resting condition in the supine position, vPWV had an average value across all subjects of $2.2 \pm 0.5$ m/s, and the CoV of the measurement was $3.6 \pm 1.1\%$. The vPWV consistently and significantly increased in all subjects during the PLR maneuver and returned towards the basal level afterwards, as shown by the box plots of Figure 5.

The percentage increases from baseline are reported in Table 1.

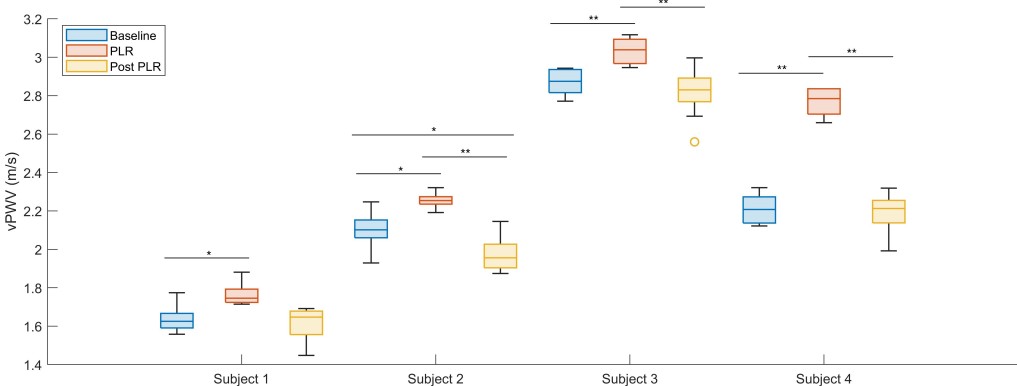

**Figure 5.** Distributions of vPWV values measured pre-, during, and post-PLR, distinctly presented for the 4 subjects by means of boxplots. The line inside of each box represents the median, while the top and bottom edges are the upper and lower quartiles, respectively. Whiskers extend over the whole data range, except for the data beyond 1.5 times the interquartile range (IRQ) above the upper quartile or below the lower quartile, which are individually plotted as circles. No data were excluded as outliers, i.e., beyond 3SD from the mean. There was consistent increase in vPWV during PLR. Differences between distributions were assessed via t-tests, adjusting resulting p-values for multiple comparisons by means of Bonferroni correction (*: $p < 0.05$; **: $p < 0.01$).

**Table 1.** Percentage increases in vPWV during PLR with respect to baseline for each subject.

| Subject | Percentage Increase |
|---------|---------------------|
| 1 | $7.6 \pm 4.1\%$ |
| 2 | $6.9 \pm 2.2\%$ |
| 3 | $5.7 \pm 2.4\%$ |
| 4 | $25.3 \pm 3.4\%$ |

Figure 6 shows the time course of vPWV during the experimental session in one subject, and compares the values obtained by the RPi-based prototype and by the original PC-based system. The relative percentage difference distribution between the two was $-3.22 \pm 0.67\%$ (average of all collected values, from all subjects). The negative value indicates that RPi generally measures a lower velocity value than that of the PC-based system.

All collected vPWV data are available as supplementary materials.

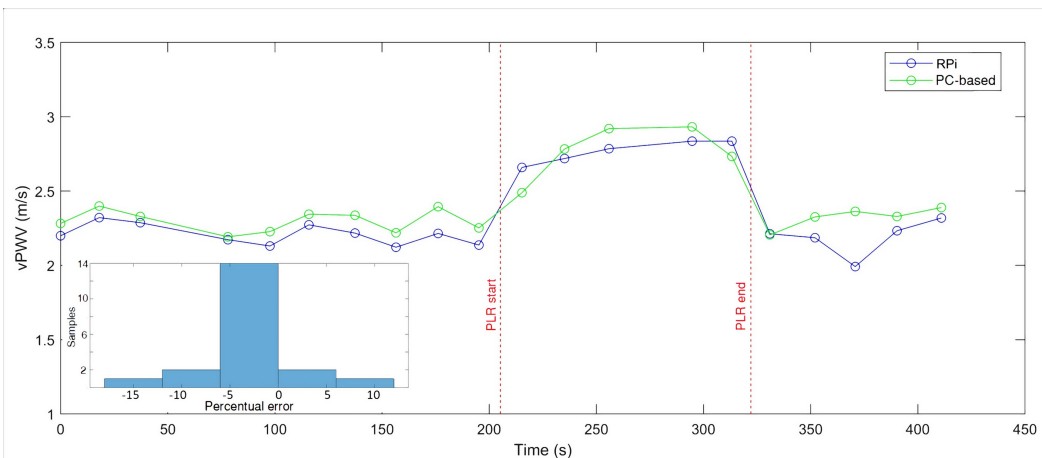

**Figure 6.** Time course of vPWV during the experimental session in one subject. Values yielded by RPi-based prototype (blue) compared with those returned by original PC-based system (green).

## 5. Discussion

For the first time, a prototype for the measurement of pulse wave velocity in the venous compartment was presented and validated. The device integrates all necessary electrical and pneumatic features, while remaining compact, battery powered, and with floating ground. This features make it electrically safe and suitable to be used on patients in hospital wards. The validation test demonstrated that the quality of the measurement is comparable with previous reports based on a personal computer and laboratory instrumentation [2,3], achieving good reliability (CoV < 4%) and sensitivity. In fact, in all 4 tested subjects, vPWV in the arm was significantly affected by a simulated fluid challenge: the PLR. This manoeuvre, performed from the supine position, is considered to displace about 150 ml of blood from the legs to the upper body, with an ensuing increase in peripheral venous pressure of less than 1 mmHg [3].

Over the last two decades, technological advancement and lower costs have led to the widespread use of single board computer platforms, such as Raspberry Pi or Jetson Nano, in a variety of fields. In particular, Raspberry Pi platform was employed in several applications in the field of biological and physiological research [8], e.g., for impedance cardiography measurement [9] and for monitoring the anaesthetic status of a patient [10]. Indeed, Raspberry Pi offers a wide range of board connectivity, custom plug-ins and interfaces that allow to develop low-cost solutions. However, a limitation of the RPi is the lack of analogue-to-digital converters. In fact, all devices that integrate RPi with the purpose of signal acquisition need an A/D converter and some circuits for signal conditioning and amplification. The A/D converter can be either on a system on a chip (SoC) [9], as in our case, or directly embedded in a microcontroller [10]. The add-on A/D board that we adopted had problems in simultaneously acquiring from multiple channels, and we observed the appearance of sampling noise at the highest sampling frequencies. We could, however, achieve satisfactory performance by sampling one analogue channel at the time and by limiting the sampling rate to 7.5 kHz. Therefore, we decided not to simultaneously acquire all signals as in the PC-based system, but to switch from one channel to another with the correct timing. Switching channel time delays were negligible and dod not affect the measurement. More significant delays occur during the handling of two simultaneous processes: the output of the cuff inflation digital signal and the acquisition of the Doppler shift signal. Those delays are probably responsible for the small difference observed in vPWV values estimated by RPi and the PC-based system.

Passing digital data from A/D converter device to RPi can use different communication protocols; the most common are SPI and Inter Integrated Circuit ($I^2C$). Both are serial protocols based on a master–slave architecture. $I^2C$ can handle more than one master at the same time and offers the advantage of easy addition of any new slaves [9,10]. SPI, on the other hand, presents a more restrictive architecture, but offers higher speed of communi-

cation [9]. In the same project, both protocols can be used for different purposes. In this approach, it is possible to exploit the advantages of both and adapt to the communication protocol offered by the A/D converter that best suits the needs [9]. In our case, the A/D plug-in board was equipped with an SPI bus.

In addition, RPi provides convenient features concerning the possibility of autonomous data processing as well as wireless communication with a remote PC for input of working settings and data display and storage. In our case, this functionality was used to manage the measurement process via GUI from a wirelessly connected computer. However, this relevant feature offers much wider possibilities. In fact, many examples of RPi-based Internet of Things (IoT) systems can be found in the literature [11,12]. RPi can be programmed to send data to a remote web database via WiFi or ethernet, adding the MAC address of the board to an IoT website. Some of these works combined the single board computer with sensors to collect some simple monitoring signals (temperature, respiratory signal, ECG, and humidity) and embedded it in an IoT system for implementing a home care monitoring strategy.

The pneumatic circuit transmis a sharp compressive stimulus in order for the generated pressure wave to be clearly identifiable. Expensive professional machines are generally adopted to the purpose of eliciting rapid tissue compressions [13]. We faced this need when investigating the muscular and skin reactivity to compressive stimuli [14,15], and developed a custom system on the basis of PC-driven valves for controlled rapid inflation and deflation of the cuff [14,16], as recently reviewed [17]. For the present application, a single 3-way valve was adopted for both inflation and deflation. The valve iswasdirectly mounted on top of the device and could conveniently be connected to the compressed air line, available in most hospitals.

Different approaches have been proposed for noninvasively assessing venous pressure or volume status that are mostly based on the echographic monitoring of the size and pulsatility of large and central veins, such as the inferior vena cava [1,18] and the jugular vein [19,20]. However, they all suffer of important limitations. Locating the point of collapse of the jugular vein is complicated by the pulsatile nature of jugular venous pressure and the necessity to estimate the hydrostatic gradient between this point and the right atrium. A rough indication about volaemic status of the patient may also be obtained by the assessment of respirophasic changes in size of the inferior vena cava [1]. This assessment, however, was criticised in the literature [21] for the low specificity [22] due to many confounding factors, among which the irregularity of spontaneous respiration [7,23] and the respiration-related displacements of the vena cava affect echographic measurements [24,25], even though methodological improvements were proposed [26–28]. A noninvasive assessment of peripheral venous pressure was also proposed based on detecting the point of collapse of superficial veins in limbs when compressed by an externally applied pressure [29], which however is limited to superficial visible veins. The present methodology overcomes most of the disadvantages of the mentioned techniques. It concerns the peripheral venous compartment (limbs) but may focus on the blood stream of large veins, which are likely to play a more relevant role in the systemic haemodynamics than that of small superficial veins. Moreover, thanks to the implemented synchronism with respiratory and cardiac activity, vPWV measurement was virtually unaffected by their intrinsic variability in amplitude and frequency.

The lack of a dedicated device has likely hindered the research in this field for many years: to our knowledge, no studies on vPWV have been published in the last 40 years in spite of the early promising results [4,5]. The prototype presented here paves the way to clinical studies that are necessary to verify the usefulness of vPWV as a possible index of peripheral vascular filling or of venosclerosis.

*Limitations*

The implementation of the device requires the use of a Doppler ultrasound for the proximal detection of the pulse wave, which has the disadvantage of been expensive, and

of requiring some experience for positioning and holding the probe in place. Moreover, the transit time of the pulse wave is measured starting from the time of triggering the cuff inflation, and thus erroneously includes the electro-pneumatic delay that precedes the actual generation of the intravascular pulse wave. This could slightly affect the accuracy of the measurement, in terms of a slight underestimation of the vPWV value. Future implementations may address these limitations. In particular, other methodologies may possibly be adopted for the pulse wave detection (e.g., tonometry [6,30,31]), which may provide a cheaper and simpler solution then Doppler ultrasound. Such an approach would also allow for a two-point detection of the pulse wave, thus eliminating the problem of the electropneumatic delay. Alternatively, acquisition and reference to the rise of air pressure in the cuff (rather than to the electrical command to the valve) could also contribute to increase the accuracy of the measurement.

The exogenous pressure pulse in venous blood may stretch the veins much over their current physiological condition, thus affecting their stiffness and consequently the vPWV measurement itself. Although vPWV could still serve as a useful vascular filling index, as suggested by early and recent investigations [2–4], the issue deserves to be elucidated by dedicated investigations, as the variability in the stimulus magnitude could introduce variability in the measurement and decrease its reliability.

## 6. Conclusions

The first compact and battery-powered device for vPWV measurement was presented and validated, demonstrating good reliability and good sensitivity to simulated fluid challenges. The device allows for extending the investigation of vPWV from healthy subjects in a laboratory to patients in hospital wards. This is necessary to initiate clinical studies oriented to understand the potential of this new parameter in characterizing the haemodynamic condition of the patient and in supporting the management of fluid therapies.

## 7. Patents

An Italian patent concerning the present methodology for vPWV measurement was deposited by University of Torino and Politecnico of Torino (BIT22246).

**Supplementary Materials:** The following are available online at https://www.mdpi.com/article/10.3390/app12042173/s1, Table S1: vPWV values from Raspberry Pi-based system and PC-based system acquired on the 4 subjects.

**Author Contributions:** conceptualization, L.E., C.F. and S.R.; methodology, A.B., L.E., R.P. and S.R.; software, A.B. and L.E.; validation, A.B., L.E. and R.P.; formal analysis, A.B.; investigation, A.B.; resources, S.R.; data curation, A.B.; writing—original draft preparation, A.B.; writing—review and editing, A.B., L.E., C.F. and S.R.; supervision, L.E. and S.R.; funding acquisition, S.R. All authors have read and agreed to the published version of the manuscript.

**Funding:** A fellowship (A.B.) was covered by a donation from Magda Passatore.

**Institutional Review Board Statement:** The study was conducted according to the guidelines of the Declaration of Helsinki, and approved by the Institutional Review Board (or Ethics Committee) of the University of Torino (23 March 2015).

**Informed Consent Statement:** Informed consent was obtained from all subjects involved in the study.

**Data Availability Statement:** The Python code used to control the device is available at https://github.com/leonardoermini/vPWV_on_RPi.git (accessed on 19 January 2022), while the experimental data presented in this study are available as Supplementary Materials.

**Acknowledgments:** The authors thank Daiana Billia for her help in developing the first version of the code used to control the A/D board as part of her Master's thesis in biomedical engineering.

**Conflicts of Interest:** A patent concerning the assessment of venous pulse wave velocity was deposited by University of Torino and Politecnico of Torino. The funders had no role in the design of the study; in the collection, analyses, or interpretation of data; in the writing of the manuscript, or in the decision to publish the results.

**Abbreviations**

The following abbreviations are used in this manuscript:

| | |
|---|---|
| A/D | Analog to digital |
| ADC | Analog to digital converter |
| CoV | Coefficient of variation |
| ECG | Electrocardiogram |
| GUI | Graphical user interface |
| I$^2$C | Inter integrated circuit |
| IoT | Internet of Things |
| MAC | Media access control |
| PC | Personal computer |
| PLR | Passive leg raising |
| RPi | Raspberry Pi |
| SoC | System on a chip |
| SPI | Serial peripheral interface |
| STD | Standard deviation |
| vPWV | Venous pulse wave velocity |

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
