# Peer review of "A Portable Device for the Measurement of Venous Pulse Wave Velocity"

_applsci, doi:10.3390/app12042173_

Round 1
Reviewer 1 Report
The authors have done a great job with this manuscript. I truly enjoyed the reading of it, its properly set up and results are clearly presented. I also think its very creative and I appreciate the section on the limitations of the system.
I have only two question/remarks, to keep it simple on the first 2 sentences of the abstract:
- First sentence: "...reconsidered as a potential index for vascular filling."
This is a very broad statement, vascular filling as indicator for what or under what circumstances? I would suggest to add at that first line one or 2 medical examples why it would be important to quantify vascular filling. It also better for the purpose of the paper. - Second sentence: "...the compressions is delivered synchronously with both heart and respiratory activity". Due to the absence of any significant pulse in the veins, a pulse has to be supplied mechanically outside the body. That is clear. Why does it have to be synchronized with the heart? Due to the absence of a pulse I would assume you can use any reasonable frequency. Did the authors test this and what goes worng? If the authors could answer this question, I would be very happy and for sure learned something new.
In short: great work, great paper.
Reviewer 2 Report
It would be nice to add a few scientific arguments quantitatively linking the parameter under discussion - venous pulse wave velocity with other physical and diagnostic parameters, for example, with the elasticity of the walls of the veins (parameters, included, in particular, in the Moens–Korteweg equation).
Reviewer 3 Report
This paper proposes a portable device for measuring venous pulse wave velocity (vpwv). The flow rate of venous blood is detected by external pressure, and the reliability of vpwv detection is verified by specific human body experiments. The device takes Raspberry PI microcomputer as the core, and integrates modules such as sampling of respiratory, ECG and ultrasonic data, wireless data interaction with host computer, control of pressure on-off valve and so on. It can quickly obtain venous wave velocity in a portable way.
Deficiencies and doubts:
1, The venous pulse is weak, so it is difficult to be located accurately. In this paper, the basilic vein is measured. I wonder how the authors select the measuring point to avoid the influence of nearby ulnar artery and brachial artery.
2, In the human body experiments, the authors introduce that passive leg raising (PLR) will lead to the increase of venous wave velocity. After the action is released, the wave speed will be slightly lower than the initial state (line 189). The author should briefly introduce the medical principles behind this result in order to enhance the reliability and persuasion of the experimental results.
3, In the discussion, the authors introduce that the AD acquisition card used will produce noise during multi-channel sampling, so they adopt single channel sampling (218-221 lines). However, the device needs to obtain three kinds of data: respiration, ECG and ultrasonic pulse at the same time. How do the authors realize the sampling of the three? If the sampling method of three-way switching is adopted, will the switching time delay affect the results?
4, The authors should introduce the data processing method in Figure 5 in more detail, especially what are the meanings in the box diagram? How are abnormal data points (such as the circle below the post PLR box diagram of the third subject in the figure) defined? How to handle and interpret these outliers?
5, There is doubt about the principle of measuring venous wave velocity: the authors believe that the normal flow of venous blood is too weak, so a large amount of venous blood is forced to flow by artificial pressurization to improve the measurement effect. Obviously, this method has changed the flow mode of venous blood. At this time, the venous blood is forced flow rather than natural flow. Can the wave velocity obtained in this way be used to describe the wave velocity of venous pulse? The authors briefly describe this measurement method in the introduction part (lines 28-41), but the references 2 and 3 cited here are also the research work of the author's team. Can the author list more convincing medical background or literature citations to show the feasibility of this measurement scheme?
6, If this method is indeed feasible, the measurement process of this method is still in doubt. The authors did not show more measurement results in this paper, and it can only refer to the previous work, that is, reference 3. If possible, the authors are requested to give longer measurement results in this paper and compare the results of forced flow with that of normal flow, so as to prove the advantages of artificial forced flow.
7, The authors explained at line 117 that the limbs were pressurized with compressed air of 2 bar (about 2 Standard atmospheres, 1500 mmHg). Is this pressure too high and out of safety range?
Round 2
Reviewer 3 Report
The authors acknowledge the limitations of the measurement method used in this paper, and explanations are given in the Limitations section. All the questions have been clearly answered and revisions have been done.